# Dicationic Bis-Pyridinium Hydrazone-Based Amphiphiles Encompassing Fluorinated Counteranions: Synthesis, Characterization, TGA-DSC, and DFT Investigations

**DOI:** 10.3390/molecules27082492

**Published:** 2022-04-12

**Authors:** Ateyatallah Aljuhani, Nadjet Rezki, Salsabeel Al-Sodies, Mouslim Messali, Gamal M. S. ElShafei, Mohamed Hagar, Mohamed R. Aouad

**Affiliations:** 1Department of Chemistry, Faculty of Science, Taibah University, Al-Madinah Al-Munawarah 30002, Saudi Arabia; ateyatallah@hotmail.com (A.A.); s7l_88@hotmail.com (S.A.-S.); aboutasnim@yahoo.fr (M.M.); 2Chemistry Department, Faculty of Science, Ain Shams University, Cairo 11566, Egypt; gamal.elshafei@sci.asu.edu.eg; 3Department of Chemistry, Faculty of Science, Alexandria University, Alexandria 21321, Egypt; mohamedhaggar@gmail.com

**Keywords:** pyridinium hydrazones, dicationic, amphiphiles, TGA-DSC study, DFT

## Abstract

Quaternization and metathesis approaches were used to successfully design and synthesize the targeted dicationic bis-dipyridinium hydrazones carrying long alkyl side chain extending from C8 to C18 as countercation, and attracted to halide (**I^-^**) or fluorinated ion (PF_6_^-^, BF_4_^-^, CF_3_COO^-^) as counteranion. Spectroscopic characterization using NMR and mass spectroscopy was used to establish the structures of the formed compounds. In addition, their thermal properties were investigated utilizing thermogravimetric analyses (TGA), and differential scanning calorimetry (DSC). The thermal study illustrated that regardless of the alkyl group length (Cn) or the attracted anions, the thermograms of the tested derivatives are composed of three stages. The mode of thermal decomposition demonstrates the important roles of both anion and alkyl chain length. Longer chain length results in greater van der Waals forces; meanwhile, with anions of low nucleophilicity, it could also decrease the intramolecular electrostatic interaction, which leads to an overall interaction decrease and lower thermal stability. The DFT theoretical calculations have been carried out to investigate the thermal stability in terms of the T_onset_. The results revealed that the type of the counteranion and chain length had a substantial impact on thermal stability, which was presumably related to the degree of intermolecular interactions. However, the DFT results illustrated that there is no dominant parameter affecting the thermal stability, but rather a cumulative effect of many factors of different extents.

## 1. Introduction

Ionic salts are entities constructed entirely of ions connected by electrostatic forces, termed ionic bonding. Ionic liquids are liquids containing ions that are referred to as molten salts; one of the main properties of these ionic compounds is low melting point, usually being liquid at room temperatures. These types of compounds have received considerable interest as tunable scaffolds over conventional molecules due to the possibility of combining one cation with several anions and vice versa, as well as zwitterionic formulations made up of a variety of cation/anion combinations [1,2]. These classes of ionic combinations are easy to synthesize and apply to a wide scope of uses [3] in the fields of organic synthesis [4], biochemistry [5], separation technology [6], electrochemistry [7], and catalytic reactions [8], which inspires researchers to design and synthesize various ionic salts due to their broad potential structural features [9]. Moreover, the dicationic ionic liquids (ILs) as salts are a new category that are gaining popularity as fascinating molecules. They became more tunable to be considered in a broader framework of numerous applied and scientific investigations [10,11,12,13,14]. This is because of their distinct physical and chemical properties from those of monocationic ILs, such as low volatility, remarkable recyclability, high thermal stability, and a relatively high decomposition temperature (T_onset_) [15,16,17,18,19]. The high-temperature uses may be one of the most diversified properties as solvents for organic reactions at high temperatures [20]. Understanding thermal stability of ionic liquids is important for the safe storage and transportation of these compounds. Considering the popularity of dicationic ILs in high-temperature applications, the results of a thermal degradation study can be helpful for using them optimally, determining their changes at high temperatures, and providing information on possible structural modifications to further improve stability. As a result, it appears essential to observe the thermal stability of the dicationic salts as unique classes of organic molecules.

The search for new dications, in addition to structural modifications of specific ions, is crucial for fine-tuning, particularly, their thermal properties. The thermal properties of ions containing imidazolium, pyrrolidinium, ammonium, and morpholinium-type cations have been studied [21,22,23,24,25,26,27,28,29,30,31]. However, to the best of our knowledge, there is no systematic investigation of the physicochemical properties of dipyridinium cations. Despite the functionalized dipyridinium cations’ promising potential, they remain unexplored in comparison to their pyridinium analogs [32]. This holds true for both their practical uses and physicochemical properties, as well as how they change as a function of chemical structure.

As a continuation of our interest in the synthesis and investigation of such dicationic ions [33,34,35,36,37,38,39,40,41,42,43], we report in the current work the design, synthesis, and investigation of the thermal properties of an array of bis-pyridinium hydrazone-based amphiphiles encompassing fluorinated counteranions.

## 2. Results and Discussion

### 2.1. Synthesis and Characterization

Considering the reason that these unique salts possess superior physicochemical properties over conventional organic molecules, we were encouraged to synthesize some novel pyridinium ions containing long alkyl chains as counter cations and different anions as illustrated in Figure 1. The most straightforward way to deliver ionic material is direct quaternization of sp^2^ nitrogen containing molecules yielding the quaternary nitrogen compounds as counter cation attracted to the halide as counter anion. The halide anion was then exchanged with a different metal anion to form the task-specific ionic liquids via the metathesis reaction [44].

For this purpose, we have planned the synthesis of novel dicationic bis-pyridinium hydrazones encompassing amphiphilic long chain tethers, starting from bis-pyridine hydrazone **1**, which was prepared as reported in the literature [45]. Initially, we performed thermal alkylation of bis-pyridine hydrazone **1** with two equivalents of the appropriate alkyl iodide with long carbon chains ranging from C8 to C18, furnishing on the formation of the desired dicationic pyridinium hydrazones **9**–**15**, tethering lipophilic side chain as counter cation and iodide as counter anion (Figure 1).

The structures of the resulted compounds **9**–**15** were illustrated based on their spectral data. Their IR spectra revealed new significant absorption bands at 2844 and 2988 cm^−1^, attributed to the aliphatic hydrogen groups confirming the incorporation of the alkyl side chains in their structures and supporting the quaternization reaction. In addition, the imine (C=N) and amino groups (-NH-) were also recorded between 1633–1688 cm^−1^ and 3400–3407 cm^−1^, respectively (Appendix A). Their ^1^H NMR spectra exhibited distinguishing multiplets and triplets around δ_H_ 4.59–4.72 ppm and δ_H_ 0.84–0.85 ppm, associated with the NC**H_2_** and C**H_3_** protons, respectively, supporting the success of the quaternization reaction. Additional methylene protons (C**H_2_**) were observed in the upfield area and were assigned to the long alkyl side chain. Similarly, the ^13^C NMR data also revealed signals from δ_C_ 60.94 to 61.61 ppm and δ_C_ 14.43 to 14.45 ppm, associated with the N-methylene (N**C**H_2_) and methyl (**C**H_3_) carbons, respectively. Additional carbon signals resonated at the aliphatic region belonging to the remaining methylene carbons, confirming the incorporation of the long alkyl chain on the pyridinium nitrogen atoms. All the remaining carbons were recorded in their respective areas (Appendix A and see experimental section).

A metathetical reaction was adopted to develop a novel series of amphiphilic dicationic pyridinium hydrazones with fluorinated anion tethers, as depicted in Figure 2. Thus, the displacement of the iodide anion was carried out through the treatment of compounds **9**–**15** with some selected fluorinated metal salts, namely potassium hexafluorophosphate (KPF_6_), sodium tetrafluoroborate (NaBF_4_), and/or sodium trifluoroacetate (NaCF_3_COO), in refluxing acetonitrile, yielding a library of amphiphilic dicationic pyridinium salts bearing hydrazone linkage and fluorinated anion.

The success of the metathesis reaction was confirmed by several spectral experiments.

It was observed that the proton and carbon signals of the resulted salts **16**–**36** were extremely similar compared to their halogenated analogues **9**–**15**, which explains why no change was recorded on their ^1^H and ^13^C NMR spectra.

Thus, the structural significance of the metathetical products **16**–**36** versus their halogenated parents **9**–**15** was established based on their ^19^F, ^31^P, ^11^B NMR, and mass spectroscopy to confirm the presence of fluorinated anions in their structures (Appendix A). Consequently, the structure of compounds **16**, **19**, **22**, **25**, **28**, **31**, and **34** tethering the hexafluorophosphate (PF_6_^-^) was evidenced by the ^31^P and ^19^F NMR spectra. Hence, the presence of a distinct multiplet in their ^31^P NMR spectra between δ_P_ (−153.02) and (−135.84) ppm complied with their proposed structures. In addition, the resonance of a new doublet between δ_F_ (−69.19) and (−69.18) ppm in the ^19^F NMR for the same derivative confirmed the presence of six-fluorine atoms in such anions (PF_6_). The presence of the BF_4_^-^ anion in the structure of the resulted dicationic pyridinium hydrazones **17**, **20, 23, 26**, **29, 32**, and **35** was supported by the investigation of their ^11^B and ^19^F NMR spectral data. Therefore, diagnostic multiplet between δ_B_(−1.39) and (−1.23) ppm was recorded in their ^11^B NMR spectra revealing the presence of the boron atom in its BF_4_ form. In addition, their ^19^F NMR spectra supported such structures, whereby exhibiting two doublets at δ_F_ (−148.21) and (−148.13) ppm.

The architectural elucidation of compounds **18**, **21**, **24**, **27**, **30**, **33**, and **36** based trifluoroacetate (CF_3_COO^-^) was supported by the examination of their ^19^F NMR spectra. The presence of a characteristic singlet resonating from (−73.61) to (−73.50) ppm confirms the incorporation of the trifluoroacetate anion in such structures.

### 2.2. Effects of Different Anions and Cations on the Thermal Stability

This study tracks the thermal stability of different synthesized amphiphilic dicationic bis-pyridinium ions, tethering different anions while especially in relation to their structural design.

#### 2.2.1. Thermogravimetric Analysis (TGA)

The short-term stability experiments give rise to a point of thermal degradation, which is occasionally termed T_decomp_. The temperature at which the sample begins to lose mass is known as the start temperature (T_start_). The thermal decomposition of sample is determined from the onset temperature (T_onset_) of TGA (thermogravimetric) curve or the peak temperature (T_max_) or (T_peak_) of the DTG (derivative thermogravimetric) curve [8]. The onset temperature is generally identified by the tangent method, and usually, different parameters follow the order T_start_ < T_onset_ < T_peak_ for the same ionic molecules.

Regardless of the alkyl group length (C_8_-C_18_) and the four anions based in the structure of the resulting molecules, the thermograms of the tested derivatives were composed of three stages in the temperature regions: 25–190, 200–450, and 450–700 °C. The first temperature range could be attributed to physically adsorbed water. The molecules that are immiscible with water tend to adsorb water from the atmosphere. On the basis of IR studies, water molecules adsorbed from the air are mostly present in a free state [46]. A major part of mass loss due to decomposition relates to the middle temperature range, as reported in Table 1. The third range corresponds to the evaporation of any residual decomposition products. It is obvious from Table 1 that there is a large difference between the melting points of the investigated compounds and their previously reported analogues [47]. The present series contains linear compounds which permit a high degree of intermolecular forces; however, the previously reported ones are angular geometrical structures that are not suitable for close-packed positioning of the molecules. Moreover, the linear structure of the present compounds also enhances the charge separation to illustrate more dipole moments. These findings could be a good explanation for the large difference in the melting points.

Figure 1 presents the thermograms of the synthesized dipyridinium bearing C8 as alkyl side chain as countercation and tethering iodide (9), hexafluorophosphate (16), tetrafluoroborate (17), and trifluoroacetate (18) anions, indicating that changing the anion results in minute differences in the thermal response of tested pyridinium salts.

On closer inspection of the corresponding DTG curves, we could trace the present differences in thermal behaviors of different compounds containing different anions, as illustrated in Figure 2.

It can be observed that the major decomposition region in the different studied molecules is composed of more than one weight loss step. It is notable that for all the tested compounds with same anion, the T_peak_ (T_max_) value does not demonstrate significant variability (Table 1).

Certain derivatives are known to thermally break down because of the anion’s nucleophilic or basic attack on the cation, or because of the anion’s early degradation into a reactive species that then reacts with the cation. Ionics containing [BF_4_^-^] and [PF_6_^-^] anions are also susceptible to hydrolysis, resulting in the release of highly corrosive hydrogen fluoride, HF. Thermal decomposition of pyridinium-based salts was proven to be caused by dealkylation of the cation via an SN^2^ reaction. The IL 1-butylpyridinium tetrafluoroborate ([bpy^+^][BF_4_^-^]), for example, decomposes into pyridine, butyl fluoride, and BF_3_ [48,49,50]. Nucleophilic and highly proton-abstracting anions, such as iodide, cause IL to decompose at much lower temperatures. Referring to the start temperature, after excluding the region due to physically adsorbed water (Table 1, Figure 3), we can arrange the tested series as **16** > **17** > **9** ≈ **18** regarding their thermal stability. The type of anion affects the thermal stability of this series following the order **PF_6_^-^** > **BF_4_^-^** > **I^-^** ≈ **CF_3_COO^-^**.

The decomposition temperatures depend primarily on the coordinating nature of the anion, with the T_decomp_ being lower for the highly coordinating anions. Such order of thermal stability was reported in cases of other derivatives with different cations [51,52]. The beginning of thermal decomposition appeared to decrease as the anion hydrophilicity was increased [53]. The strength of H-bonding between anion and water increases in the order **PF_6_^-^** < **BF_4_^-^** < **CF_3_COO^-^** [46]. The values of weight loss in the region of physically adsorbed water are in line with the following order (Table 1): as the strength of adsorbed water increases, the loss on thermal treatment decreases. It is notable that the values of T_onset_ do not verify that sequence of thermal stability, which agrees with the general implication that T_onset_ overestimates the thermal stability. Furthermore, the melting point demonstrated no correlation with stability or molecular weight, however, it demonstrated the lowest difference between melting point and T_start_ in case of the most stable ion (Table 1).

For the C9-pyridinium series, the CF_3_COO^-^ containing salt had lower stability than the two containing PF_6_^-^ and BF_4_^-^ based on the value of T_start_, Table 1.

The thermogram of pyridinium ions containing C_9_ as a side chain and CF_3_COO^-^ as a counter anion, 21 in Figure 4, demonstrates that the weight loss in the temperature 25–700 °C terminated at 60%, indicating incomplete degradation. This illustrates that numerous separate thermal degradation processes occur in several temperature regimes, and that a nonvolatile solid is created. Mass losses < 100%, indicating solid residue formation, likely through polymerization of the anions or their thermal decomposition products [54]. No plausible explanation can currently be offered to account for the behavior of the bis-pyridinium **21**.

Different figures of the investigated bis-pyridinium ions bearing different alkyl side chains varying from C10 to C18 are presented in the Appendix A. While the product compassing the C10 alkyl chain and fluorinated anions (**22**–**24**) exhibited higher thermal stability than their iodide analogue **11**, the derivative containing the **PF_6_^-^** as anion (22) exhibited an abnormally low melting point, lower than T_start_. This sample started melting before starting weight loss due to decomposition. The order of thermal stability according to the effect of anion hydrophilicity is retained in the pattern of C12 (**12** and **25**–**27**) (Table 1). For the series compassing the C14 chain, the four compounds (**13** and **28**–**30**) demonstrated comparable thermal stability with somewhat high and comparable percentages of physically adsorbed water with an extendable temperature range. On the other hand, the C16 and C18 series (**14**, **15**, and **31**–**36**) demonstrated comparable percentages of physically adsorbed water, but lower values compared to the C14 analogues, and both series illustrated a comparatively lower range of T_start_ values. Increasing the alkyl chain length decreases the thermal stability; this is observed for imidazolium ionic derivatives with basic [Cl] [14], and weakly coordinating BF_4_^-^ and/or PF_6_^-^ anions [55,56]**.** This is in accordance with the calculated activation energies for the dealkylation reaction which decreases upon elongation of the alkyl chain [22]. The influence of chain length on thermal stability with different cations and anions was interpreted as follows: while a longer chain length results in greater van der Waals forces, it could also decrease the intramolecular electrostatic interaction, which leads to an overall decreased interaction and lower thermal stability [57,58].

#### 2.2.2. Differential Scanning Calorimetry (DSC)

Some samples were studied using DSC to detect the effect of both the cation and the anion on the thermal degradation of the pyridinium salts under study. The series of different alkyl groups with iodide anion was selected to highlight the role of the alkyl chain length in thermal decomposition. The pyridinium salts of chain length C9 and C14 with three different anions were tested to check the anion effect on the thermal decomposition. Figure 5 presents the DSC curve of sample C14 (I^-^) superposed on the corresponding DTG curve to illustrate the details potentially gained from a DSC study. The similar illustrations for other samples are gathered in the Appendix A.

The DSC curve illustrates the endothermic processes due to loss of physically adsorbed water up to 100 °C, indicating a different strength of adsorption since interaction energies of water with anions are larger than with cations [59]. The endothermic effect near 200 °C is due to melting; owing to the strong interaction between the cation and the anion in the dicationic compound, many dicationic salts generate melting points higher than 100 °C [60]. The major decomposition process expressed itself in the strong endotherm appearing in the temperature range 250–300 °C.

##### Role of Alkyl Chain

Due to the minute differences in the profiles of the DSC curves of the samples with iodide as anion and different alkyl chain lengths, the curves are presented in two (Figure 6a,b) one for C8–C12 chains, the second for C14–C18 chains.

For all studied samples, the endothermic effects in the region of physically adsorbed water up to ~115 °C imply its presence with variable extents of interaction, as inferred from the appearance of more than one effect in most cases. While it was reported that in mono-cationic ionic liquids the water-cation interaction strength diminishes with increasing alkyl chain length of the cation [14], the results in Figure 6 do not completely comply with this observation if the temperature of the endothermic effect is considered. Alternatively, in dicationic pyridinium salts, the situation does not appear straightforward to detect simply.

Furthermore, for mono-cationic ionic liquids, the chain length of the alkyl group on the cation was proven by some researchers to have no large effect on the thermal stability of the ILs [13,61,62]. Conversely, others have reported a decrease in thermal stability upon increasing the alkyl chain length [14,55,56]. Considering that ionic liquids have an endothermic breakdown mechanism, probably by loss of an alkyl chain, the results in Figure 6 demonstrate that the major endothermic effect due to decomposition in the temperature region above 250 °C appears independent from alkyl chain length in the group C8-C12. Conversely, decomposition temperature (endotherm peak) increases from 265 to 278 °C when the alkyl length changed from C14 to C18. Another difference is the presence, after the melting endotherm, of a small endothermic effect before the major one in samples C8-C12 that is lacking in all the samples with higher alkyl length. This effect is most likely due to the increasing stability of linear, aliphatic carbo-cations and/or free radicals, with their increasing chain length, which makes them better leaving groups during the heating and, thus, promotes the breakdown of the C–N bond [52,63].

##### Role of Anion

The DSC curves for C9 and C14 samples with three different anions, mainly (I**^-^**), (PF_6_**^-^**), and (BF_4_**^-^**), are presented in Figure 7.

For C9 series, the temperatures of the endotherms due to desorption of physically adsorbed water demonstrate the order of anion hydrophilicity I_-_
**>** BF_4-_
**>** PF_6-_, while in the case of C14 series, the hydrophobicity of the cation has dominance over the variability of anions.

As in the case of I^-^, increasing the alkyl chain length from C9 to C14 in the samples with BF_4_^-^ and PF_6_^-^ anions demonstrated change in the mode of endothermic decomposition from multiple to single effect. However, the compound with iodide as anion is observably the least stable among the three examined samples as inferred from the appearance of only two endothermic effects, compared to three in other anions. Furthermore, in the latter cases, the endothermic effects illustrated greater tendency to extend toward higher temperatures placing the analogue with PF_6_^-^ anion as the most stable among the three samples. This observation is more evident in case of C14 samples.

### 2.3. DFT Theoretical Calculations

DFT theoretical computations were done at the base stand set B3LYP 6-311g (d,p) for selected examples of the dicationic bis-pyridinium hydrazones **9**–**36**. The proposed compounds’ optimal geometrical structures were computed in gas phase using Gaussian 9. To determine the lowest-energy geometrical structure, all compounds were minimized and optimized, including a structural optimization estimation for each molecule. The optimization technique was used to identify the geometrical structure for the lowest-energy conformations, in which the atoms, bond lengths, bond angles of the compounds were displayed until a new lowest-energy geometrical structure was produced, which is known as convergence. The optimized structures were then utilized to calculate the frequency and some essential thermodynamic properties. All optimized molecular structures of all substances have been confirmed to be stable due to the lack of the imaginary frequency, as illustrated in Figure 8 of derivatives **9** and **16**–**18** tethering the same chain length C-8 and different counter anion, and set of derivatives **17**, **20**, **23**, **26**, **29**, and **32** encompassing the same counterion BF_4_ and different chain lengths. The calculated thermal and dimension parameters of the synthesized ions were predicted using DFT utilizing the same technique and base set, and the results are reported in Table 2. These parameters have been used to illustrate the obtained thermal stability in terms of the T_onset_ results.

As illustrated in Figure 9, the dipole moment of the compound is highly affected by the type of counteranion. The derivatives bearing BF_4_^-^ and CF_3_COO^-^ anions demonstrated the highest dipole moment and the iodide demonstrated the least. Although the dipole moment could be expected as one of the important parameters that may affect the thermal stability of the compound, Figure 9 does not reveal regular correlation of the calculated dipole moment with the Tonset.

On the other hand, we have investigated the effect of the calculated dimensions with respect to the T_onset_ as an indicator of the thermal stability. Clearly, the iodide derivative displayed the longer length and the triflouroacetate displayed the shortest. The length and the width of the compounds could be affecting the degree of thermal stability by inducing a degree of van der Waals forces. The regular dependence of the T_onset_ on the length of the compounds, as the length decreases the T_onset_ value decreases, except for the PF_6_^-^ derivative, which could be affected by another factor (Figure 10).

Moreover, Figure 11 illustrates the correlation between the T_onset_ with calculated energy of the prepared compounds **9** and **16**–**18** of the same chain length C-8 and different counter anions. As demonstrated in Figure 11, hexafluorophosphate derivative has the least energy with the highest stability over the other counter anion derivatives. The stability of the compound could be a good explanation for the corresponding thermal stability, whereby the higher the internal energy, the lower the thermal stability of the compounds and the lower the T_onset_. The other derivatives demonstrated an irregular dependence of the T_onset_ on the internal energy; this result could illustrate that the T_onset_ is affected by many factors to different extents.

Conversely, the effect of the chain length has also been investigated by calculating the thermal parameters and the dimensions of the prepared compounds of different chain length of the same counter anion. It could be inferred from Figure 12 that the length and width of the compounds affect the thermal stability in a way that has been discussed before; the longer length and the larger width could illustrate the degree of stability because of the higher intermolecular interactions. As the length and width increase, the T_onset_ increases up to C-12, then a constant value until C-16, then it decreases again at C-18.

Finally, one of the important factors that could affect thermal stability is the aspect ratio: the ratio of the length to the width of the compounds. As the aspect ratio increases, more backing of the compounds could be permitted and consequently, the thermal stability could be enhanced. The irregular dependence of the T_onset_ on the length of the chain in C-18 (Figure 13) could be explained by the lower value of the aspect ratio of C-18 derivative.

## 3. Experimental Section

### 3.1. General Procedures for Synthesis of Amphiphilic Dicationic Pyridinium Hydrazone with Iodide Counter Anions **9**–**15**

Bis-pyridine hydrazone **1** (1 mmol) in acetonitrile (30 mL) was refluxed for 6–12 h with the appropriate long alkyl iodide **2**–**8** with a carbon chain ranging from C-8 to C-18 (2.2 mmol). TLC was used to track the progress of the reactions. The solvent was greatly reduced by evaporation under reduced pressure, and the precipitate produced was collected by filtration, washed with acetonitrile, dried, and crystallized from ethanol to yield the halogenated dicationic bis-pyridinium derivatives **9**–**15**.

### 3.2. General Procedures for Synthesis of Amphiphilic Dicationic Bis-Pyridinium Hydrazone with Fluorinated Counter Anions **16**–**36**

In acetonitrile, a mixture of dicationic liquids **9**–**15** (1 mmol) and metal salts potassium hexafluorophosphate (KPF_6_), sodium tetrafluoroborate (NaBF_4_), and/or sodium trifluoroacetate (NaCF_3_COO) (2.5 mmol) were refluxed for 16 h. After cooling, filtration was used to collect the resulting precipitate, which was then washed with acetonitrile, dried, and crystallized from ethanol to produce the desired dicationic bis-pyridinium derivatives **16**–**36**.

NB: The characterization of the compounds is given in Appendix A.

### 3.3. Computational Details

The quantum chemical calculations of the studied compounds were carried out by using the DFT method with the B3LYP functional and 6–311G (d,p) basis set by Gaussian 09 software. The maximum optimization of geometries was done by minimizing the energies corresponding to all the geometrical parameters without changing any molecular symmetry constraints. Gauss View 5.8 was used to visualize and draw the frontier molecular orbitals as well as optimize the structure. Calculations of the frequency indicate the absence of any imaginary frequency modes, which proved the minimum energy of the optimized structures. The gauge including atomic orbital (GIAO) method was done to determine NMR calculations with the same level of theory, and the 1H isotropic tensors were used as a reference to the TMS calculation at the same level.

## 4. Conclusions

A focused library of dicationic bis-dipyridinium hydrazones carrying long aliphatic side chains ranging from C8 to C18 as countercation, and attracted to halide and/or fluorinated ion as counteranion was successfully synthesized and characterized using different spectroscopic experiments.

A thermal stability investigation demonstrated that the thermograms of the tested derivatives have three stages, regardless of the length of the alkyl group (Cn) for the four anions. Moreover, longer chains of the investigated compounds demonstrated higher van der Waals forces, but they may also reduce intramolecular electrostatic interaction, resulting in an overall drop in interaction and thermal stability. DSC and TG results indicated the thermal stability of synthesized dicationic ionic liquids to decrease as the nucleophilicity of the anion increased. Thermal decomposition proceeds endothermically with a dealkylation process, demonstrating a distinct dependence on the alkyl chain length.

Additionally, the DFT theoretical results revealed that the type of counteranion and chain length had a substantial impact on thermal stability. The degree of intermolecular interactions has been attributed for these findings according to several relationships between the experimental and theoretical data. Conversely, the DFT results revealed that there is no single dominant parameter impacting thermal stability, but rather an accumulative effect of multiple parameters of varying degrees.

## Data Availability

Not applicable.

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
