# Peer review of "Dicationic Bis-Pyridinium Hydrazone-Based Amphiphiles Encompassing Fluorinated Counteranions: Synthesis, Characterization, TGA-DSC, and DFT Investigations"

_molecules, 2022, doi:10.3390/molecules27082492_

Round 1
Reviewer 1 Report
The opinions given in the first round of reviews have not been fundamentally changed with the revisions which have been made to the paper.
The novelty and performance of these species as ionic liquids is still difficult to understand in comparison to well established compounds.
Author Response
Response to Reviewers’ Comments
Ref. No.: Molecules-1619504
Title: Dicationic bis-pyridinium hydrazone-based amphiphiles encompassing fluorinated counter anions: Synthesis, Characterization, TGA-DSC and DFT Investigations
Journal: Molecules
Dear Editor/ reviewers,
We would like to convey our sincere gratitude to you and respected reviewers for their valuable comments to improve the manuscript. The manuscript has been revised substantially as suggested. We have tried our best to follow the reviewers’ suggestion.
The following actions were performed in the revised version of the manuscript. The corrections are highlighted in red color in the attached revised version of the manuscript. Here, we also have added below the answers next to the queries raised by the reviewers.
Reply to reviewer 1:
- The opinions given in the first round of reviews have not been fundamentally changed with the revisions which have been made to the paper.
The novelty and performance of these species as ionic liquids is still difficult to understand in comparison to well established compounds.
Response : Thermal behavior and structure are of significant considerations in the pharmaceutical industry because they are regularly monitored and have the potential to impact drug processing and bioavailability. Our interest in the antitumorigenic activities of the investigated compounds, we have shown in previous studies, made the thermal and theoretical aspects of prime importance to be considered as presented in this study.
As shown above, the authors' responses to the editor and reviewer comments improved the current manuscript. As a result, we must thank the reviewers for reading and commenting on the manuscript. Finally, please do not hesitate to contact me if you have any further questions.
Best regards
Dr. Prof. Mohamed Reda Aouad
Professor of Organic Chemistry
Reviewer 2 Report
The manuscript is significantly improved. The authors have addressed the comments satisfactorily and, therefore I support the publication of the revised manuscript
Author Response
Response to Reviewers’ Comments
Ref. No.: Molecules-1619504
Title: Dicationic bis-pyridinium hydrazone-based amphiphiles encompassing fluorinated counter anions: Synthesis, Characterization, TGA-DSC and DFT Investigations
Journal: Molecules
Dear Editor/ reviewers,
We would like to convey our sincere gratitude to you and respected reviewers for their valuable comments to improve the manuscript. The manuscript has been revised substantially as suggested. We have tried our best to follow the reviewers’ suggestion.
The following actions were performed in the revised version of the manuscript. The corrections are highlighted in red color in the attached revised version of the manuscript. Here, we also have added below the answers next to the queries raised by the reviewers.
Reply to reviewer 2:
The manuscript is significantly improved. The authors have addressed the comments satisfactorily and, therefore I support the publication of the revised manuscript.
Response : Thank you for your helpful advice.
As shown above, the authors' responses to the editor and reviewer comments improved the current manuscript. As a result, we must thank the reviewers for reading and commenting on the manuscript. Finally, please do not hesitate to contact me if you have any further questions.
Best regards
Dr. Prof. Mohamed Reda Aouad
Professor of Organic Chemistry
Reviewer 3 Report
The manuscript entitled Dicationic bis-pyridinium hydrazone-based amphiphiles encompassing fluorinated counter 2 anions: Synthesis, Characterization, TGA-DSC and DFT Investigations has interesting results regarding synthesis, characterization and thermal properties of dicatonic bis-pyridinium hydrazone-based compounds with different counter anions. However, the manuscript needs to be improved to be suitable for publication.
1) The authors should revise the introduction section, and improve it. There is a considerable lack of coherence in the text. It is very challenging for the reader as there are many long sentences without text flow.
For example, it is not clear what the authors wanted to say in “Some ions are referred to as molten salts which are a sort of liquids that contain ions called ionic liquids…” Moreover, one of the main properties of these sorts of ionic compounds is low melting point i.e. they are usually liquid at room temperatures. I think it is important to emphasize this.
2) What are the purities of synthesized ionic liquids? Did the authors perform a purification operation after obtaining ionic liquids?
3) The authors should use Tstart and Tend in Table 1 as in the text.
4) Figures of IR spectra are missing in SI.
5) lines 108-109 ”with some selected fluorinated metal salts, namely hexafluorophosphate (KPF6), sodium tetrafluororoborate (NaBF4), and/or sodium trifluoroacetate (NaCF3COO), in refluxing…”
Add potassium in front of hexafluorophosphate and correct spelling for “tetrafluororoborate” in tetrafluoroborate (also in line 410).
6) lines 293, 339, etc. Use Figure X instead of figure X. There are both versions through the text.
Author Response
Response to Reviewers’ Comments
Ref. No.: Molecules-1619504
Title: Dicationic bis-pyridinium hydrazone-based amphiphiles encompassing fluorinated counter anions: Synthesis, Characterization, TGA-DSC and DFT Investigations
Journal: Molecules
Dear Editor/ reviewers,
We would like to convey our sincere gratitude to you and respected reviewers for their valuable comments to improve the manuscript. The manuscript has been revised substantially as suggested. We have tried our best to follow the reviewers’ suggestion.
The following actions were performed in the revised version of the manuscript. The corrections are highlighted in red color in the attached revised version of the manuscript. Here, we also have added below the answers next to the queries raised by the reviewers.
Reply to reviewer 3:
The manuscript entitled Dicationic bis-pyridinium hydrazone-based amphiphiles encompassing fluorinated counter 2 anions: Synthesis, Characterization, TGA-DSC and DFT Investigations has interesting results regarding synthesis, characterization and thermal properties of dicatonic bis-pyridinium hydrazone-based compounds with different counter anions. However, the manuscript needs to be improved to be suitable for publication.
- The authors should revise the introduction section, and improve it. There is a considerable lack of coherence in the text. It is very challenging for the reader as there are many long sentences without text flow.
For example, it is not clear what the authors wanted to say in “Some ions are referred to as molten salts which are a sort of liquids that contain ions called ionic liquids…” Moreover, one of the main properties of these sorts of ionic compounds is low melting point i.e. they are usually liquid at room temperatures. I think it is important to emphasize this.
Response : This point has been addressed in the revised version (See manuscript).
- What are the purities of synthesized ionic liquids? Did the authors perform a purification operation after obtaining ionic liquids?
Response : All compounds are of higher purity, as clearly indicated by NMR and mass spectral data. All compounds were purified by crystallization from ethyl alcohol. In the experimental part, a brief description of the purification of all compounds by crystallization was added (See experimental section).
- The authors should use Tstartand Tend in Table 1 as in the text.
Response : This point has been addressed in the revised version (See manuscript).
- Figures of IR spectra are missing in SI.
Response : Some IR spectra have been included in the SI (See supplementary Information).
- lines 108-109” with some selected fluorinated metal salts, namely hexafluorophosphate (KPF6), sodium tetrafluororoborate (NaBF4), and/or sodium trifluoroacetate (NaCF3COO), in refluxing…” Add potassium in front of hexafluorophosphate and correct spelling for “tetrafluororoborate” in tetrafluoroborate (also in line 410).
Response : This point has been addressed in the revised version (See manuscript).
- lines 293, 339, etc. Use Figure X instead of figure X. There are both versions through the text.
Response : This point has been addressed in the revised version (See manuscript).
As shown above, the authors' responses to the editor and reviewer comments improved the current manuscript. As a result, we must thank the reviewers for reading and commenting on the manuscript. Finally, please do not hesitate to contact me if you have any further questions.
Best regards
Dr. Prof. Mohamed Reda Aouad
Professor of Organic Chemistry
Reviewer 4 Report
The manuscript molecules-1619504 "Dicationic bis-pyridinium hydrazone-based amphiphiles encompassing fluorinated counter anions: Synthesis, Characterization, TGA-DSC and DFT Investigations" by Mohamed Reda Aouad and co-workers describes the synthesis of a series of dicationic bis-dipyridinium hydrazones with alkyl (C8-C18) side chain and halide (I-) or fluorinated ion (PF6-, BF4-, CF3COO-) as counter anion. The thermal stability of the obtained compounds was studied by physical methods, i.e. TGA, DSC.
Questions and comments:
1) Synthesis and characterization of the compounds 9-36 described in the manuscript was published earlier in another work of the authors [Int. J. Mol. Sci. 2021, 22(19), 10487; https://doi.org/10.3390/ijms221910487 ]. Therefore, there is no novelty in this manuscript from the point of view of synthesis.
2) The physical characteristics, namely the melting points, in the previous article of the authors differ from the results given in this manuscript. How can the authors explain this?
3) Scheme 1 does not indicate the values of n. Scheme 2 - the general variant of writing compounds as 16-36 is unclear. Please change the decoding of the structures as the initial compounds are signed in Scheme 2.
4) The synthesized compounds were not dried according to the experimental part of the manuscript. It is known that impurities of a small amount of water greatly affect the properties of ionic salts, in particular ionic liquids. The amount (%) of water in the obtained ionic compounds should be estimated before planning other experiments.
5) Lines 188-189. "Thermal decomposition of pyridinium-based ILs was discovered to be caused by dealkylation of the cation via an SN2 reaction". It looks like a hypothesis. The authors should be propose a mechanism for the destruction of the compounds and prove it by physical methods.
Author Response
Response to Reviewers’ Comments
Ref. No.: Molecules-1619504
Title: Dicationic bis-pyridinium hydrazone-based amphiphiles encompassing fluorinated counter anions: Synthesis, Characterization, TGA-DSC and DFT Investigations
Journal: Molecules
Dear Editor/ reviewers,
We would like to convey our sincere gratitude to you and respected reviewers for their valuable comments to improve the manuscript. The manuscript has been revised substantially as suggested. We have tried our best to follow the reviewers’ suggestion.
The following actions were performed in the revised version of the manuscript. The corrections are highlighted in red color in the attached revised version of the manuscript. Here, we also have added below the answers next to the queries raised by the reviewers.
Reply to reviewer 4:
The manuscript molecules-1619504 "Dicationic bis-pyridinium hydrazone-based amphiphiles encompassing fluorinated counter anions: Synthesis, Characterization, TGA-DSC and DFT Investigations" by Mohamed Reda Aouad and co-workers describes the synthesis of a series of dicationic bis-dipyridinium hydrazones with alkyl (C8-C18) side chain and halide (I-) or fluorinated ion (PF6-, BF4-, CF3COO-) as counter anion. The thermal stability of the obtained compounds was studied by physical methods, i.e. TGA, DSC.
Questions and comments:
- Synthesis and characterization of the compounds 9-36described in the manuscript was published earlier in another work of the authors [ J. Mol. Sci.2021, 22(19), 10487; https://doi.org/10.3390/ijms221910487]. Therefore, there is no novelty in this manuscript from the point of view of synthesis.
Response : Thank you very much for your comment, however, the present study reports the synthesis of dicationic bispyridinium hydrazones encompassing amphiphilic long chain tethers from the condensation of isonicotinic acid hydrazide with 4-formylpyridine. In contrast, the work reported in our previous work [Int. J. Mol. Sci. 2021, 22 (19), 10487; https://doi.org/10.3390/ijms221910487] relates to the synthesis and anticancer activity of dicationic bipyridinium hydrazone analogs derived from 3-formylpyridine (not 4-formylpyridine). Furthermore, the thermal and theoretical aspects are the main objectives of this study.
- The physical characteristics, namely the melting points, in the previous article of the authors differ from the results given in this manuscript. How can the authors explain this?
Response : Thank you for your comment; however, the previously published work concentrated on the anticancer activity of analogous compounds of 3-formylpyridine. The current work is entirely different. Furthermore, the mps of the present compounds must be significantly different from those of previously reported compounds due to differences in the charge distribution and dipole moments of the 3-formyl and 4-formyl derivatives, which must affect intermolecular forces and thus the mps.
- Scheme 1 does not indicate the values of n. Scheme 2 - the general variant of writing compounds as 16-36is unclear. Please change the decoding of the structures as the initial compounds are signed in Scheme 2.
Response : As requested, the two schemes were corrected (See manuscript).
- The synthesized compounds were not dried according to the experimental part of the manuscript. It is known that impurities of a small amount of water greatly affect the properties of ionic salts, in particular ionic liquids. The amount (%) of water in the obtained ionic compounds should be estimated before planning other experiments.
Response : Because the compounds were all obtained in solid form, they were all purified by crystallization and dried (See manuscript).
- Lines 188-189. "Thermal decomposition of pyridinium-based ILs was discovered to be caused by dealkylation of the cation via an SN2reaction". It looks like a hypothesis. The authors should be propose a mechanism for the destruction of the compounds and prove it by physical methods.
Response : This point has been addressed in the revised version (See manuscript).
As shown above, the authors' responses to the editor and reviewer comments improved the current manuscript. As a result, we must thank the reviewers for reading and commenting on the manuscript. Finally, please do not hesitate to contact me if you have any further questions.
Best regards
Dr. Prof. Mohamed Reda Aouad
Professor of Organic Chemistry
Reviewer 5 Report
The manuscript entitled "Dicationic Bis-pyridinium Hydrazone-Based Amphiphiles Encompassing Fluorinated Counter Anions: Synthesis, Characterization, TGA-DSC, and DFT Investigations" describes the synthetic process and characterization of Dicationic Bis-pyridinium Hydrazones.
Here are some recommendations for revision
- The authors should include the reaction condition in the scheme.
- How the reaction progress were monitored?
- The authors should include the application of such molecules in the introduction.
- Missing previous reported similar dicationc compounds
Example:
Journal of Molecular Structure Volume 1207, 127756.
Author Response
Response to Reviewers’ Comments
Ref. No.: Molecules-1619504
Title: Dicationic bis-pyridinium hydrazone-based amphiphiles encompassing fluorinated counter anions: Synthesis, Characterization, TGA-DSC and DFT Investigations
Journal: Molecules
Dear Editor/ reviewers,
We would like to convey our sincere gratitude to you and respected reviewers for their valuable comments to improve the manuscript. The manuscript has been revised substantially as suggested. We have tried our best to follow the reviewers’ suggestion.
The following actions were performed in the revised version of the manuscript. The corrections are highlighted in red color in the attached revised version of the manuscript. Here, we also have added below the answers next to the queries raised by the reviewers.
Reply to reviewer 5:
The manuscript entitled "Dicationic Bis-pyridinium Hydrazone-Based Amphiphiles Encompassing Fluorinated Counter Anions: Synthesis, Characterization, TGA-DSC, and DFT Investigations" describes the synthetic process and characterization of Dicationic Bis-pyridinium Hydrazones.
Here are some recommendations for revision:
- The authors should include the reaction condition in the scheme.
Response : As requested, the two schemes were corrected (See manuscript).
- How the reaction progress was monitored?
Response : TLC was used to track the progress of the reactions.
- The authors should include the application of such molecules in the introduction.
Response : This point has been addressed in the revised version (See manuscript).
- Missing previous reported similar dicationic compounds
Example: Journal of Molecular Structure Volume 1207, 127756.
Response : As requested, the proposed reference [Journal of Molecular Structure Volume 1207, 127756] has been included in the revised version (See manuscript).
As shown above, the authors' responses to the editor and reviewer comments improved the current manuscript. As a result, we must thank the reviewers for reading and commenting on the manuscript. Finally, please do not hesitate to contact me if you have any further questions.
Best regards
Dr. Prof. Mohamed Reda Aouad
Professor of Organic Chemistry
Round 2
Reviewer 4 Report
I thank the authors for answering my questions and improving the manuscript.
I recommend comparing the physical properties of the compounds obtained in this manuscript with analogues from the authors' previous article (J. Mol. Sci.2021, 22(19), 10487; https://doi.org/10.3390/ijms221910487). Comparison and explanation of the difference in properties should be added to the text of the manuscript.
Author Response
Response to Reviewers’ Comments
Ref. No.: Molecules-1619504
Title: Dicationic bis-pyridinium hydrazone-based amphiphiles encompassing fluorinated counter anions: Synthesis, Characterization, TGA-DSC and DFT Investigations
Journal: Molecules
Dear Editor/ reviewer,
We would like to convey our sincere gratitude to you and respected reviewers for their valuable comments to improve the manuscript. The manuscript has been revised substantially as suggested. We have tried our best to follow the reviewers’ suggestion.
The following actions were performed in the revised version of the manuscript. The corrections are highlighted in blue color in the attached revised version of the manuscript. Here, we also have added below the answers next to the queries raised by the reviewers.
Reply to reviewer 4:
Comments and Suggestions for Authors
I thank the authors for answering my questions and improving the manuscript.
I recommend comparing the physical properties of the compounds obtained in this manuscript with analogues from the authors' previous article (J. Mol. Sci.2021, 22(19), 10487; https://doi.org/10.3390/ijms221910487). Comparison and explanation of the difference in properties should be added to the text of the manuscript.
Response : As per the suggestion of learned reviewer, this point has been addressed in the revised version; lines165 to line 172 and table (See manuscript).
As shown above, the authors' responses to the editor and reviewer comments improved the current manuscript. As a result, we must thank the reviewers for reading and commenting on the manuscript. Finally, please do not hesitate to contact me if you have any further questions.
Best regards
Dr. Prof. Mohamed Reda Aouad
Professor of Organic Chemistry
